# Typical Combined Travel Mode Choice Utility Model in Multimodal Transportation Network



**Yue Liu [1,2,3], Jun Chen [1,2,3,*], Weiguang Wu [4] and Jiao Ye [1,2,3]**

1. Jiangsu Key Laboratory of Urban ITS, Southeast University, Nanjing 211189, China; hitliuyue@163.com (Y.L.); yejiao23@163.com (J.Y.)
2. Jiangsu Province Collaborative Innovation Center of Modern Urban Traffic Technologies, Southeast University, Nanjing 211189, China
3. School of Transportation, Southeast University, Nanjing 211189, China
4. Hangzhou City Planning and Design Academy, Hangzhou 310000, China; 15751868298@163.com
* Correspondence: chenjun@seu.edu.cn; Tel.: +86-25-013-913-945-222

**Abstract:** The primary purpose of this paper is to explore the mechanism of combined travel mode choice in multimodal networks. To meet the objective, stated preference survey and revealed preference survey are designed under short, middle, and long travel distance scenarios. Data including travelers' socio-economic/personal information, trip characteristics, and mode choice are collected and analyzed. To recognize the influential factors of mode choice, a nested logit model is established. A value of time estimation and sensitivity analysis are conducted to quantify the influencing degree. The results reveal that cost has a significant influence on the short-distance travel mode; waiting time is perceived as the most important factor in short-distance scenario, and transfer-walking time as the most significant in middle and long distance scenario. Moreover, the traveler is more sensitive to the decrease of the transfer walking time than increase. Regarding socio-economic/personal information, travelers aged 40–50 prefer to choose combined travel mode than other ages; female travelers have a greater acceptance of metro-based transfer travel than male; individuals with higher economic level have a positive image of metro than bus.

**Keywords:** Travel Behavior; Stated Preference (SP) Survey; Revealed Preference (RP) Survey; Nested Logit Model; Sensitive Analysis; Value of Time (VOT)

## 1. Introduction

With the rapid development of economy and acceleration of urbanization, city regions extend continuously, and the average travel distance of residents has increased significantly. The choice of travel modes is no longer limited to single ways such as cars, buses, and metro, and it tends to be multimodal.

Due to the difference in travel speed, comfort, and travel cost of each mode, each travel mode is dominant in different travel situations. To utilize the advantages of various modes of transportation expansively, travelers often choose combined travel modes ranging from car to metro, bus to metro, bike to bus, and bike to metro. To explore the mechanism of the traveler's choice of each mode, understanding to what extent traveler's socio-economic, personal information, and trip characteristics affect the choice of combined travel mode, utility analysis of combined travel mode is significant.

Several authors have investigated the influential factors associated with mode choice. Hartgen found that individual socio-economic attributes, travel attitudes, and the modes of transportation influence the travel mode choice behavior [1]. Bhat and Srinivasan believe that households with higher income have the preference to auto mode [2]. Yang and Li found that females prefer to choose the bus

than male [3]. Travel time saving [4] and reasonable ticket fare [5] are vital factors for public transit attractiveness. Punctuality is another influential factor of mode choice [6]. For commuters, they are more willing to choose the motorized travel mode [7].

In the research field of travel utility, McFadden [8] and Ben-Akiva and Lerman [9] first put the discrete choice model into practice. The MNL model is widely used in the travel mode choice analysis because it is a simple mathematical form that enables ease of estimation and interpretation, and the ability to add or remove choice alternatives [10]. However, the MNL model has been widely criticized for its Inter-dependence of Irrelevant Alternatives [IIA] property [10]. To eliminate the IIA property, the nested logit model was developed. LO developed a three-level NL choice model to deal with the complex and inter-related decisions in a multi-modal network and examined the effect of fare competition [11]. Based on the data obtained in Barcelona, Asensio analyzed the determinants of travel mode choice for suburbanized commuters using the NL model. The results reveal that values of travel time savings are high for the commuters who use cars [12]. Using the NL model, Joachim Scheiner focused on the relationship between travel mode choice, travel distance, and city size, and the results suggest that car owners are more inclined to walk a given distance in the cities than in small towns, even more so if they live in a central urban area [13].

Most of the above studies on mode choice model lack the division of experimental scenarios based on the connection between single and combined travel modes and the decision of travel distances. Instead, all travel modes are simultaneously selected as alternatives for comparison, which do not match the actual choice behavior. The relationship between travel modes and the travel distance of each mode method can be balanced scientifically and rationally to distinguish experimental scenarios that are needed for further research.

The current research contributes to the travel mode choice model by taking features of the combined travel mode into consideration. Build mode choice models under different distance scenarios to improve the accuracy and feasibility of the NL model. By conducting VOT and sensitivity analysis, the mechanism of mode choice in multimodal network is explored.

The remainder of the paper is organized as follows: Section 2 presents the survey design and implementation, feature analysis on travel mode. Section 3 introduces the NL model and definition of variables. The results and analysis of model estimation are shown in Section 4, followed by VOT estimation and sensitive analysis in Section 5. Section 6 demonstrates the findings and suggestions of this paper.

## 2. Data Acquisition and Analysis

### 2.1. Survey Design

Revealed preference (RP) survey and stated preference (SP) survey are combined in this study. RP survey is mainly used to gather data on current behavior including personal information and trip characteristics [14], and it can reflect the actual choice of the respondent. Nevertheless, there are some limitations in the RP survey. Firstly, due to a certain degree of correlation between variables of RP data, the questionnaire may generate redundant information. Secondly, the response to diverse attributes combinations that are not observed in the market cannot be captured using this method [15]. Through the SP survey, the preference of respondents under hypothetical scenarios can be obtained, and a more extensive selection scheme can be provided for the respondents. Meanwhile, with the aid of uniform design, alternatives are generated in a low-level correlation [15–17]. The survey content is shown in Figure 1.

At first, the respondents were asked about their socio-economic and personal information, including gender, age, occupation, car ownership, and income. Then, questions about their real trip characteristics including travel mode and travel time were asked. There were also some questions about their alternative travel mode and satisfaction with service quality including comfort and reliability of multiple modes [18,19].

Next, respondents were requested to choose a scenario that they were familiar with and to complete the SP survey [20–22]. The alternatives are described by four attributes: Commute time, transfer waiting time, and transfer-walking time. The attribute levels are yielded to actual commute travel situation in Nanjing. The number of attribute levels are determined by complexity of the design and proved to be reasonable by uniform design experimentation. Uniform design form is shown in Table 1.

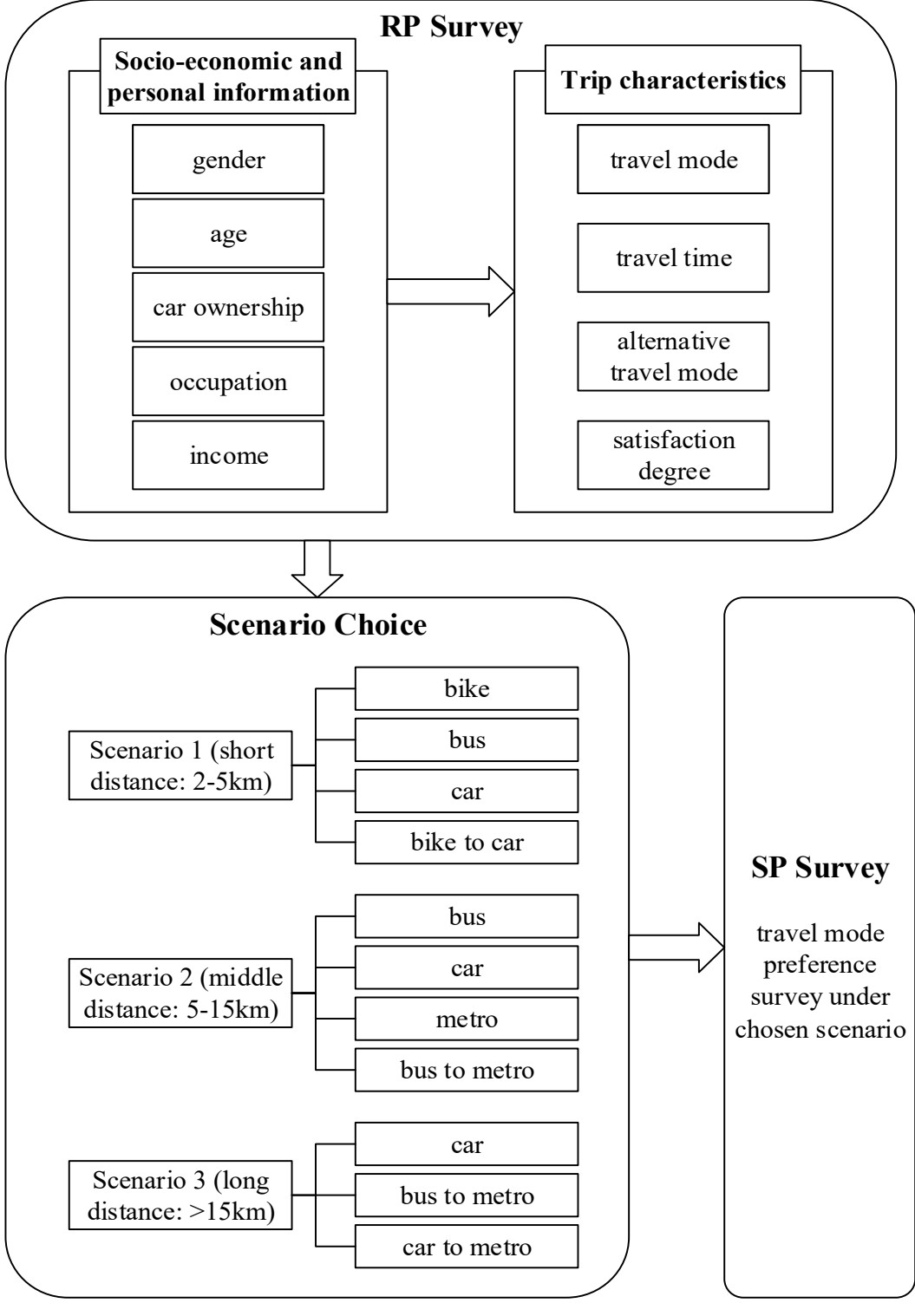

**Figure 1.** Survey Content.

**Table 1.** Attributes under Each Scenarios.

| Scenario | Travel Distance (km) | Attributes | Levels (min) | Uniform Design Form |
|---|---|---|---|---|
| 1 | 2–5 | Commute time of bus | 8,10,12,15,18,20 | $U_6^*(6^4)$ |
|  |  | Transfer waiting time | 2,5,8,10,12,15 |  |
|  |  | Commute time of bike | 2,3,4,5,6,7 |  |
|  |  | Transfer walking time | 1,2,3,4,5,6 |  |
| 2 | 5–15 | Waiting time of bus | 2,5,8,10 | $U_8^*(4^5)$ |
|  |  | Waiting time of metro | 1,2,3,5 |  |
|  |  | Commuting time of bike | 3,5,8,10 |  |
|  |  | Transfer walking time | 3,4,5,6 |  |
|  |  | Commute time of metro | 10,15,20,25 |  |
| 3 | >15 | Waiting time of bus | 2,5,8,10 | $U_8^*(4^5)$ |
|  |  | Waiting time of metro | 1,2,3,5 |  |
|  |  | Commute time of car | 10,20,30,40 |  |
|  |  | Transfer walking time | 3,4,5,6 |  |
|  |  | Commute time of metro | 20,30,40,50 |  |

* indicates a design form with better uniformity.

## 2.2. Survey Implementation

The questionnaire was mainly published in the web-based format. Its advantages are: Lower cost, quick delivery, benefit for collecting and analysis data, randomness of respondents, and ease of switching scenario. However, there are still some limitations: Multiple submissions, lack of on line experience, and instructions [22]. Thus, the on-site survey was supplemented.

The online survey lasted from November 6 to December 6, 2017, for the duration of one month. Meanwhile, the on-site survey was implemented during the morning peak hour (7:30–9:00) and evening peak hour (17:00–19:00) in December. The survey site was decided in China, Nanjing (118.5° E, 118.5° N), which is one of the biggest cities in Southeast China. Up to now, Nanjing had operated 10 metro lines, with a total length about 375 km, ranking fourth in China. It owns a multi-modal transport network including metro network, regular bus network, road system, and bike share services, which create an enabling environment for investigation. The survey spots were distributed in four metro stations, a few shopping malls, and several restaurants (see Figure 2).

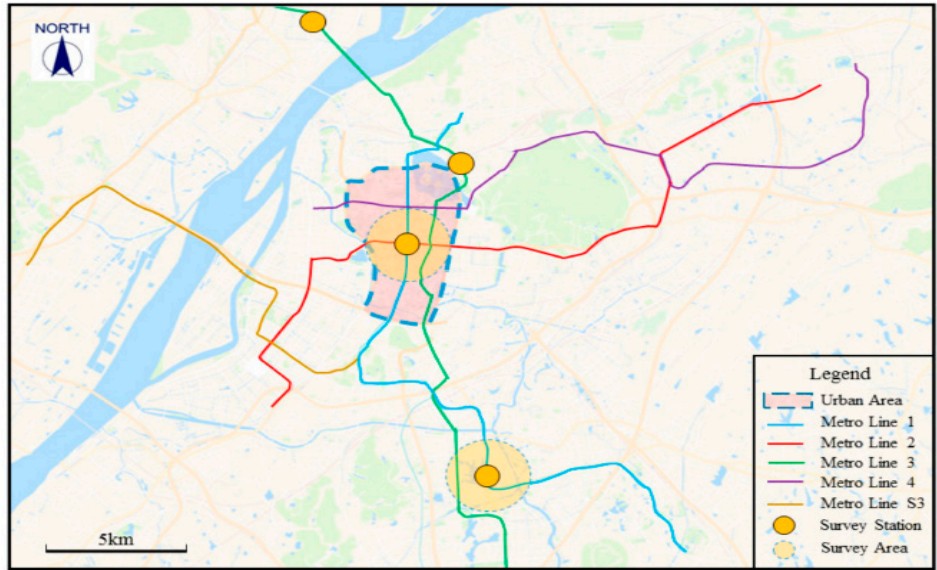

**Figure 2.** Distribution of Survey Sites.

A total of 644 questionnaire data were filled. After the preliminary elimination of multiple submission and nonresponse questionnaires, 589 valid data were retained. There are 271, 232, and 86 valid questionnaires collected under scenario 1, 2, and 3, respectively. One complete questionnaire includes six to eight experiments. Therefore, a total of 271 × 6 + 232 × 8 + 86 × 8 = 4170 travel mode choice samples were finally obtained through the SP survey. Table 2, in Section 2.3, shows that the sample meets the target requirements.

**Table 2.** Proportion of Personal and Socio-Economic Data.

| | | Total Sample (N = 589) | Scenario 1 (N = 271) | Scenario 2 (N = 232) | Scenario 3 (N = 86) |
|---|---|---|---|---|---|
| Gender | Male | 0.57 | 0.60 | 0.54 | 0.58 |
| | Female | 0.43 | 0.40 | 0.46 | 0.42 |
| Age(years) | 10–20 | 0.05 | 0.08 | 0.03 | 0.02 |
| | 20–30 | 0.43 | 0.47 | 0.43 | 0.31 |
| | 30–40 | 0.36 | 0.31 | 0.38 | 0.49 |
| | 40–50 | 0.11 | 0.07 | 0.13 | 0.14 |
| | 50 and over | 0.05 | 0.06 | 0.03 | 0.03 |
| Individual Income (yuan/month) | 3000 and less | 0.23 | 0.31 | 0.17 | 0.13 |
| | 3000–6000 | 0.25 | 0.27 | 0.27 | 0.18 |
| | 6000–10,000 | 0.27 | 0.23 | 0.33 | 0.28 |
| | 10,000–20,000 | 0.16 | 0.10 | 0.17 | 0.27 |
| | 20,000 and more | 0.09 | 0.09 | 0.07 | 0.14 |
| Number of Cars | 0 | 0.33 | 0.42 | 0.33 | 0 |
| | 1 | 0.53 | 0.47 | 0.53 | 0.77 |
| | 2 and more | 0.14 | 0.11 | 0.14 | 0.23 |

### 2.3. Traveler Personal and Socio-Economic Attribute Analysis

The data of 589 valid questionnaires were collected to obtain the personal and socio-economic attributes of travelers in different situations.

Table 2 presents the proportion of respondents with different personal and socio-economic attributes. Among the respondents, 43% were women and 57% were men. Respondents in the 20–30, 30–40, and 40–50 age groups accounted for 43%, 36%, and 11% of the total, respectively. As this paper focuses on commuting travel, the age group of respondents was consistent with the target group of the study. It can be seen that the income distribution matches the real situation. The respondents with one or more cars account for 67% of the total.

### 2.4. Travel Mode Characteristics Analysis

The travel time distribution of every single mode and the travel mode distribution under different travel distance scenarios is shown in Figures 3 and 4.

Figure 3 shows that the travel time of bike is basically less than 30 min, and the travel time of regular bus, car and metro is concentrated at 10~30 min, 30~40 min, greater than 40 min intervals, taking the proportion of 50%, 20%, and 20%, respectively. The results give evidence of the scenario set before.

As can be seen in Figure 4, bike is the primary mode in scenario 1, taking the proportion of more than 40%. Single car and bus mode are popular in scenario 1, while less than 10% commuters would like to choose the combined mode. In scenario 2, car becomes the dominate travel mode, followed by metro and bus. With the increase of travel distance, there are more commuters who are willing to choose the combined mode. Car to metro and car become the primary travel mode in scenario 3.

Figure 5 shows that the comfort evaluation of the four travel modes is consistent in the degree of congestion and physical exertion. The single bus mode has the lowest indicators. Additinoally, the single car mode has the highest indicators because of the congestion on the road.

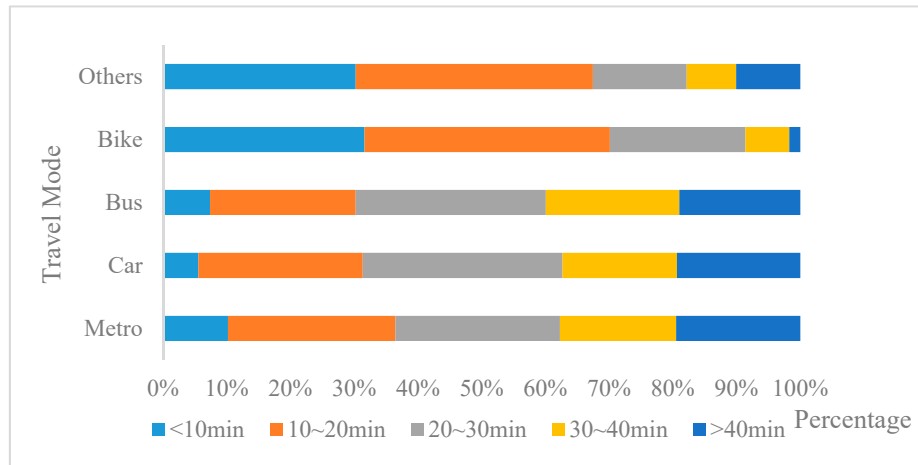

**Figure 3.** Travel Time Distribution of Single Travel Mode.

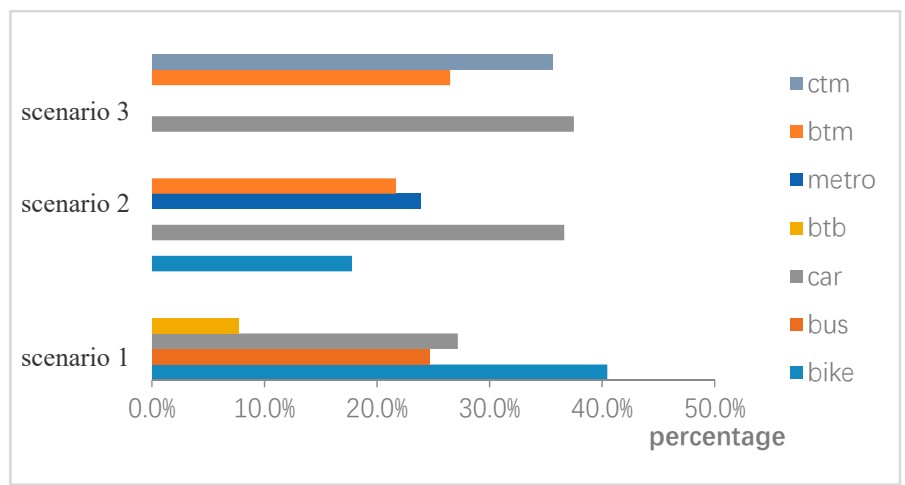

**Figure 4.** Mode Choice Distribution of Three Scenarios.

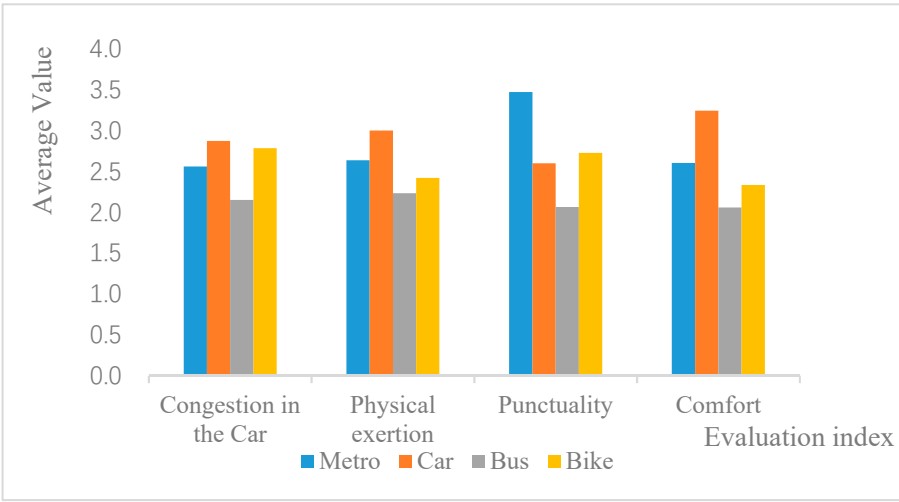

**Figure 5.** Comfort Evaluation of Single Travel Mode.

Regarding punctuality, the metro mode is the most advantageous. Due to the short travel distance and point-to-point feature, the bike mode is also punctual. At the same time, the punctuality of the car mode is low because of the congestion on the roads during the peak hours.

When it comes to comfort, the car mode has the highest indicator. The indicator of the metro mode is also high because of its spacious interior, proper ventilation, and stable operation. In the meantime, poor ventilation and moving stationary significantly reduce comfort of the bus mode. Additionally, due to the reduced level of bike lane facilities, the comfort indicator of single bike mode is low.

Through the correlation analysis of the above four indicators, the two related variables of congestion in the car and physical exertion are eliminated, and two independent variables, which are comfort and punctuality, are retained.

## 3. Methodology

### 3.1. Travel Utility

Travel utility refers to the absolute value measurement made by people in travel decisions based on time, cost, comfort, and safety factors related to travel behavior.

Travelers follow the principle of maximizing utility in the process of travel behavior selection, which means that the travel plan with the most extensive travel utility is always selected. The stochastic utility theory considers service to be a random variable, which can be expressed as the sum of the utility determination term and the random term. The formula is as follows:

$$U_{ij} = V_{ij} + \varepsilon_{ij}, \tag{1}$$

The variable $U_{ij}$ is the utility variable, it varies with the traveler $i$ and also with the travel scheme $j$; $V_{ij}$ is systematic (or representative) components; and $\varepsilon_{ij}$ is disturbance.

Random item 1 is an unmeasurable systematic error, obeying a specific function distribution (normal distribution or Gumble distribution), determining item 2 is the measurable part of the system, and there is a functional relationship between the factors affecting the travel behavior, called the utility function. Usually, the utility function takes a linear form, which is expressed as follows:

$$V_{ij} = x_{ij}\beta_{ij} + z_i\gamma_i \; (i = 1, \dots, n; j = 1, \dots, J), \tag{2}$$

The explanatory variable $x_{ij}$ is the utility attribute of the selection item $j$, which varies with the traveler $i$ and also with the travel scheme $j$; the explanatory variable $z_i$ is the traveler characteristic variable that changes only with the traveler $i$; and $\beta_{ij}$ and $\gamma_i$ are variable coefficients.

### 3.2. Nested Logit (NL) Model

This paper uses the NL (Nested-Logit) model to analyze the combined travel utility.

The NL model group allows the correlation of alternatives within the nest and keeps independence between pairs of nests [23]. Take the two-layer NL model as an example, and the nested structure is shown in Figure 6.

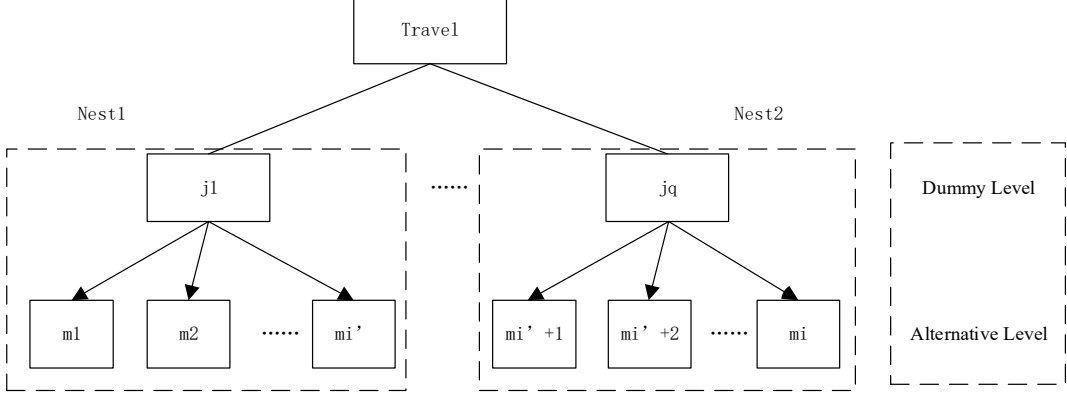

**Figure 6.** NL Nested Structure Diagram.

The probability that the traveler chooses the travel mode *m* is:

$$P_n(m) = P_n(j) \, P_n(m|j) = \frac{e^{(V_j+V_{j'})\mu^j}}{\sum_{j'\in J} e^{(V_j+V_{j'})\mu^{j'}}} \times \frac{e^{(V_{mj}+V_m)\mu^m}}{\sum_{m'\in M} e^{(V_{m'j}+V_{m'})\mu^{m'}}} \tag{3}$$

The dummy level is split to single travel mode and combined travel mode, and the alternative level is divided into specific modes. According to the different travel distances, the nested structure of the three scenarios is shown in Figure 7.

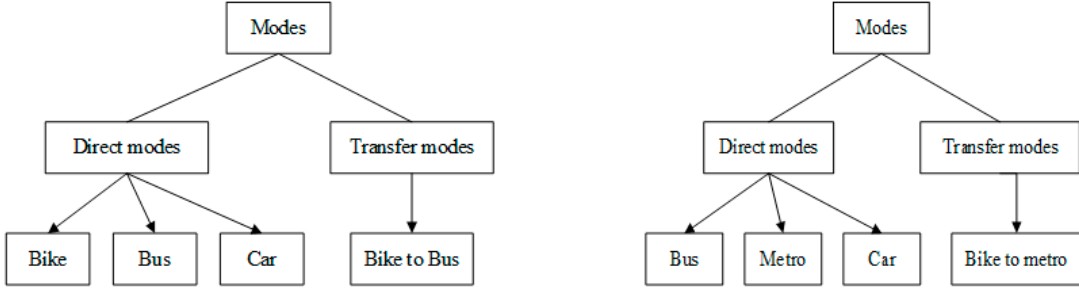

a. nested structure in short distance scenario    b. nested structure in middle-distance scenario

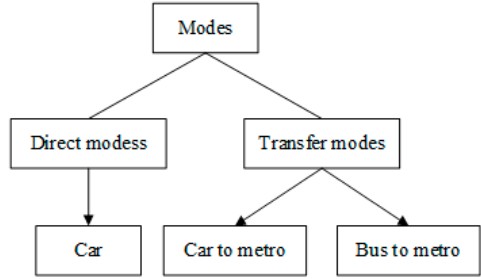

c. nested structure in long-distance scenario

**Figure 7.** Nested Structure in Three Scenarios.

### 3.3. Variables Definition

After determining the nested structure of the NL model, it is necessary to clarify the variable settings at each level. Select the factors that mainly affect the choice of single and combined travel mode as the variables of the dummy level, and select the elements that directly affect the choice of travel mode as the variables of the alternative level. Variable selection and definition are as shown in Table 3.

**Table 3.** Variable Definition.

|  | Detailed Variables | Unit | Denotation |
|---|---|---|---|
|  | Male | – | Reference |
|  | Female | – | female |
|  | 10–20: Age from 10 to 20 years old | – | age1 |
|  | 20–30: Age from 20 to 30 years old | – | age2 |
| Dummy level | 30–40: Age from 30 to 40 years old | – | age3 |
|  | 40–50: Age from 40 to 50 years old | – | age4 |
|  | 50 and over: Age over 50 years old | – | Reference |
|  | Commute distance | km | distance |
|  | Transfer walking time | min | ttwalk |
|  | Transfer waiting time | min | twait |

**Table 3.** *Cont.*

| Detailed Variables | Unit | Denotation |
|---|---|---|
| Cost | yuan [1] | cost |
| Commute time | min | time |
| Waiting time | min | wait |
| Walk to transferring bus time | min | twalk |
| 3000 and below: Income from 0 to 3000 | yuan/month | income1 |
| 3000–6000: Income from 3000 to 6000 | yuan/month | income2 |
| 6000–10,000: Income from 6000 to 10,000 | yuan/month | income3 |
| 10,000–20,000: Income from 10,000 to 20,000 | yuan/month | income4 |
| 20,000 and more: Income over 20,000 | yuan/month | Reference |
| 0: Possess no car | – | nveh1 |
| 1: Possess one car | – | nveh2 |
| 2 and more: Possess more than two cars | – | Reference |
| Satisfaction with the punctuality of one mode | – | pun |
| Satisfaction with the comfort of one mode | – | com |

(with "Alternative level" as the row label spanning the data rows.)

[1] The unit of RMB.

## 4. Model Estimation and Results

### 4.1. Estimation Result

Using the STATA software [24,25], combined with the traveler's socio-economic and personal information data in the SP survey and the travel mode choice under the short, middle, and long travel scenarios, the parameter estimation results for the NL model is shown in Table 4.

**Table 4.** NL Model Parameter Estimation Table.

|  |  | Variables | Short Distance | | Middle Distance | | Long Distance | |
|---|---|---|---|---|---|---|---|---|
|  |  |  | Coefficient | P | Coefficient | P | Coefficient | P |
| Dummy level | Personal variables | **Reference** |  |  |  |  |  |  |
|  |  | gender | – |  | 0.54 | 0.000 | 0.47 | 0.030 |
|  |  | age1 | – |  | – |  | – |  |
|  |  | age2 | 1.17 | 0.002 | −0.72 | 0.000 | – |  |
|  |  | age3 | 1.01 | 0.013 | – |  | −1.63 | 0.000 |
|  |  | age4 | 1.51 | 0.001 | 0.56 | 0.001 | −1.45 | 0.000 |
|  |  | **Reference** |  |  |  |  |  |  |
|  | Cost and time variables | distance | −0.22 | 0.007 | – |  | −0.27 | 0.000 |
|  |  | twalk | – |  |  |  | – |  |
|  |  | ttwalk | – |  | −0.36 | 0.000 | −0.41 | 0.001 |
|  |  | twait0 | −0.12 | 0.055 | −0.25 | 0.000 | – |  |
| Alternative level | Socio-economic variables | income1_bike | 2.03 | 0.003 | – |  | – |  |
|  |  | income1_metro | – |  | -0.35 | 0.028 | – |  |
|  |  | income1_btb | 1.71 | 0.001 | – |  | – |  |
|  |  | income2_btb | 0.79 | 0.015 | – |  | – |  |
|  |  | income3_bus | −1.41 | 0.058 | – |  | – |  |
|  |  | income3_ctm | – |  | – |  | 1.82 | 0.004 |
|  |  | income3_btm | – |  | 0.26 | 0.058 | −2.52 | 0.009 |
|  |  | income4_bus | −3.94 | 0.000 | – |  | – |  |
|  |  | **Reference** |  |  |  |  |  |  |
|  |  | nveh1_bike | −2.39 | 0.000 | – |  | – |  |
|  |  | nveh1_bus | −1.72 | 0.001 | −0.29 | 0.010 | – |  |
|  |  | nveh2_metro | – |  | 0.32 | 0.010 | – |  |
|  |  | nveh2_btm | – |  | 1.82 | 0.000 | – |  |
|  |  | **Reference** |  |  |  |  |  |  |

Table 4. *Cont.*

|  |  | Variables | Short Distance | | Middle Distance | | Long Distance | |
|---|---|---|---|---|---|---|---|---|
|  |  |  | Coefficient | P | Coefficient | P | Coefficient | P |
| Cost and time variables |  | cost | −1.12 | 0.000 | −0.06 | 0.006 | −0.46 | 0.000 |
|  |  | twalk | − |  | −0.05 | 0.004 | −0.48 | 0.058 |
|  |  | time | −0.08 | 0.051 | −0.02 | 0.030 | −0.07 | 0.099 |
|  |  | twait1 | −0.38 | 0.002 | − |  | − |  |
|  |  | tinveh | − |  | − |  | −0.05 | 0.052 |
| Alternative level | Comfort and punctuality | carpun_bike | 1.75 | 0.000 | − |  |  |  |
|  |  | carpun_btb | 1.22 | 0.000 | − |  |  |  |
|  |  | carpun_ctm | − |  | − |  | −1.13 | 0.008 |
|  |  | carpun_btm | − |  | − |  | 2.59 | 0.009 |
|  |  | bikepun_bike | −2.23 | 0.000 | − |  | − |  |
|  |  | bikepun_bus | − |  | 0.21 | 0.027 | − |  |
|  |  | bikepun_metro | − |  | 0.16 | 0.018 | − |  |
|  |  | bikepun_btm | − |  | −0.67 | 0.000 | − |  |
|  |  | bikepun_btb | −1.40 |  | − | 0.012 | − |  |
|  |  | buspun_btm | − |  | − | 0.000 | −1.63 | 0.000 |
|  |  | metropun_ctm | − |  | − | 0.027 | 3.96 | 0.008 |
|  |  | metropun_btm | − |  | − | 0.036 | 4.96 | 0.000 |
|  |  | carcom_ctm | − |  | − |  | 1.24 | 0.000 |
|  |  | carcom_btm | − |  | − |  | −1.95 | 0.002 |
|  |  | buscom_bike | −1.28 |  | − |  | − |  |
|  |  | buscom_bus | −1.66 |  | −0.29 |  | − |  |
|  |  | buscom_ctm | − |  | − |  | 1.69 | 0.000 |
|  |  | buscom_btm | − |  | 0.30 |  | −3.00 | 0.000 |
|  |  | buscom_btb | −0.98 |  | − |  | − |  |
|  |  | metrocom_metro | − |  | −0.13 |  | − |  |
|  |  | metrocom_btm | − |  | −0.16 |  | −1.04 | 0.001 |
|  |  | bikecom_bike | −1.69 |  | − |  | − |  |
|  |  | bikecom_btb | −0.53 |  | − |  | − |  |
| Adjusted R2 |  |  | 0.211 | | 0.157 | | 0.287 | |
| VOT(yuan/h) |  |  | 9.13 | | 20 | | 4.29 | |

*4.2. Result Analysis*

4.2.1. Socio-Economic and Personal Variables

The coefficients of personal income indicate that individuals with low income (below 6000 yuan) have the preference for bike and bus mode. High income earners (higher than 6000 yuan) avoid to use bus mode. Interestingly, there is no negative effect on metro mode or car to metro. Moreover, a high number of vehicles have a negative effect on bike and bus mode in scenario 1 and scenario 2. Again, it has no negative influence on the metro mode. The results may suggest that individuals with high economic level may perceive traveling with the bus mode as not matching their socio-economic level and have a more positive impression on metro than bus. The finding is consistent with Bhat at al. (2006), who found that individuals from high income-earning households are likely to use rail transit than the bus or non-motorized travel modes [26].

The gender parameter is positive in sign and significant statistically, which reveals that, compared with male, female travelers have a stronger ability to accept the combined travel modes. Another study of gender effect on travel mode choice (John and He, 2017) shows that female is more likely to use public transportation than male [27].

Travelers aged 40–50 prefer to choose combined travel mode than other ages.

4.2.2. Cost and Time Variables

In line with the traditional cognition, travel cost, and time have a negative influence on utility. However, there are some new findings by dividing the distance scenarios. The cost has different

extents of effects on three scenarios, most significant in scenario 1, followed by scenario 3 and scenario 2. The result may supply implications for price making policy.

Travel time, waiting time, transfer walking time and in vehicle time are perceived heterogeneously in three scenarios. In scenario 1, waiting time for bus is perceived great importance (followed by travel time). Other time variables are not significant. With the increasing of travel distance, transfer walking time become significant to travelers in scenario 2 and scenario 3, and in-vehicle time became significant in scenario 3. The results are understandable, since the increasing travel distance will enlarge the perception of transfer walking time. Unlike Iseki and Taylor (2009) who states that waiting times are more valued than walking times, we found that it varies with travel distance. In order to explore the reason, further discussion will be conducted in next Section.

### 4.2.3. Comfort and Punctuality

The coefficients of comfort and punctuality reveals that high punctuality of metro is the most important factor to attract commuters from car mode to combined mode.

## 5. Discussion

### 5.1. Variation of VOT by Distance Interval

The aim of this section is to demonstrate the differences in the VOT (value of time) associated with distance intervals. VOT can be defined as a traveler's willingness to pay for travel time saving [28]. Several approaches were used to estimate the VOT. The mode choice model is employed in this paper.

According to the behavioral value theory, the VOT is obtained by this formula:

$$VOT = \frac{\partial V}{\partial t} \Big/ \frac{\partial V}{\partial \gamma} = \frac{c}{b} \tag{4}$$

The *V* parameter is a utility function, *t* is time, $\gamma$ parameter is cost. The *c* and *b* are the coefficients of time and cost in the utility function.

Combining the results of NL model with formula (4), the VOT under different intervals is shown in Figure 8 [25]. It can be seen that VOT in middle distance has the highest value, nearly two times of that in the long distance scenario, while the VOT in the short distance is the lowest. Another research paper (Zong et al., 2009), in Jilin, China, divided two distance scenarios, found that VOT increase when the trip distance increases [29]. The difference may due to the property of area and the deviation of travel distance intervals.

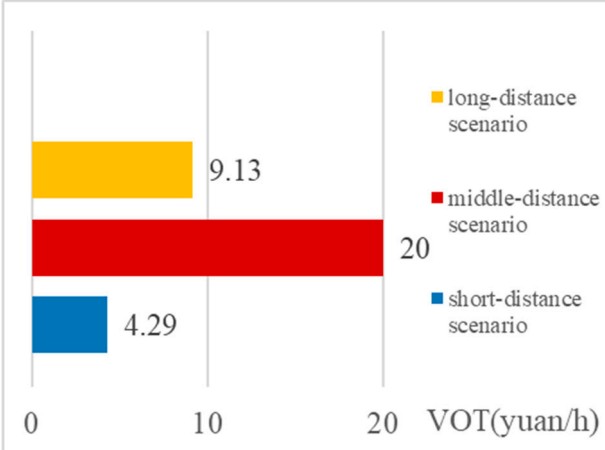

**Figure 8.** VOT under different distance scenarios.

Figure 9 shows the variation of VOT by multiple time of combined mode. The results may give support to the result analysis in Section 4.2. For mode bike to bus, waiting time is more valued than transfer waiting time. For mode bus to metro and car to metro, transfer walking time is severely perceived. Therefore, utilizing the design of facilities to reduce transfer walking time is critical to attract long-distance commute to combined mode from auto mode.

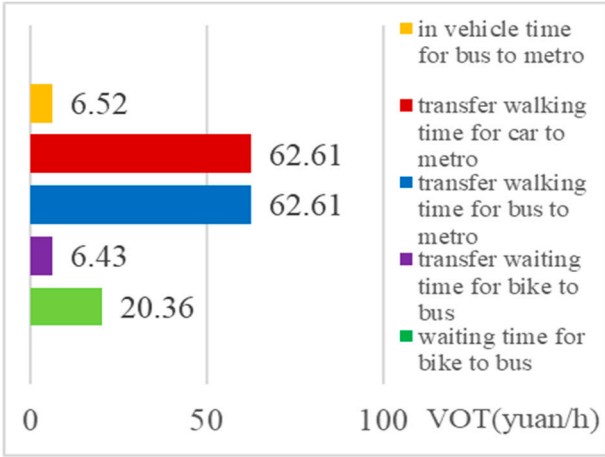

**Figure 9.** VOT of combined travel mode.

*5.2. Sensitivity Analysis*

After the analysis above, we find that setting reasonable price and reducing transfer walking time are incentive strategies for commuters to choose combined mode. Thus, quantifying the change of mode choice probability with the variation of specific factor is necessary.

In this section, sensitive analysis is selected to meet the purpose. We concentrate the analysis on two factors: Travel cost and transfer walking time. Since these two factors are both significant in scenario 2, and the VOT of this scenario is highest, so scenario 2 is taken as an example. The results are shown as follows.

Table 5 present the changes in travel mode choice probability with the 10% increase (and decrease) of travel cost. Commuters are more sensitive to cost of car and metro. After raising the cost of car, car-choice probability decreases by 3.99%, metro-choice probability increases by 3.34%, btm increases by 0.64%. Adjusting metro fares also presents an obvious effect.

**Table 5.** Cost Sensitivity Analysis Table.

| Change of Cost | | Car Choice Probability | Bus Choice Probability | Metro Choice Probability | btm Choice Probability |
|---|---|---|---|---|---|
| Before change of cost | | 21.12% | 0.40% | 65.90% | 12.57% |
| Cost of car | +10% | 17.13% | 0.42% | 69.24% | 13.21% |
| Cost of car | −10% | 25.76% | 0.38% | 62.03% | 11.84% |
| Cost of bus | +10% | 21.14% | 0.31% | 65.96% | 12.59% |
| Cost of bus | −10% | 21.10% | 0.52% | 65.82% | 12.56% |
| Cost of metro | +10% | 24.86% | 0.48% | 59.86% | 14.80% |
| Cost of metro | −10% | 17.68% | 0.34% | 71.46% | 10.52% |
| Cost of btm | +10% | 21.74% | 0.42% | 67.85% | 9.99% |
| Cost of btm | −10% | 20.36% | 0.39% | 63.54% | 15.71% |

Table 6 present the changes of the travel mode choice probability with the 10%, 30, and 50% increase (and decrease) of transfer walking time.

**Table 6.** Transfer Walking Time Sensitivity Analysis.

| Change of ttwalk | Car Choice Probability | Bus Choice Probability | Metro Choice Probability | btm Choice Probability |
|---|---|---|---|---|
| ttwalk −10% | 20.17% | 0.39% | 62.94% | 16.50% |
| ttwalk −30% | 17.59% | 0.34% | 54.89% | 27.18% |
| ttwalk −50% | 14.17% | 0.27% | 44.21% | 41.35% |
| ttwalk 0% | 21.12% | 0.40% | 65.90% | 12.57% |
| ttwalk +10% | 21.87% | 0.42% | 68.24% | 9.47% |
| ttwalk +30% | 22.89% | 0.44% | 71.42% | 5.25% |
| ttwalk +50% | 23.47% | 0.45% | 73.23% | 2.85% |

Table 6 shows that when the transfer walking time increases by 10% (30%, 50%), the probability of bike to the metro changes from 12.57% to 9.47% (5.25%, 2.85%), and when the transfer walking time decreases by 10% (30 %, 50%), the probability of bike to the metro becomes 16.5% (27.18%, 41.35%), showing that travelers are more sensitive to the reduction of transfer time than decrease. The results suggest that even slight reduction of transfer walking time works on promoting the choice probability of the combined mode.

## 6. Conclusions

This paper takes exploring the mechanism of combined travel in case of commuting trips overall purpose, conduct NL model establishment, VOT estimation, and sensitive analysis using combined SP-RP approach.

The survey was carried out in Nanjing, China. According to the data analysis of questionnaire, dominate travel mode in different travel intervals was recognized. Bike is the primary mode in scenario 1, while less than 10% commuters would like to choose the combined mode. Car is the dominate travel mode in scenario 2. With the increase of travel distance, there are more commuters who are reluctant to choose combined mode. car to metro and car become the primary travel mode in scenario 3. Characteristics of each mode including travel time distribution and comfort evaluation were also analyzed and compared in this part, with the respect of getting an overall cognition of the collected data.

Next, NL model was employed to recognized the key factors influencing the mode choice. After estimation of the modeling results, we conclude the factors into three parts: Socio-economic and personal variables, travel cost and travel time, and comfort and punctuality. We found that individuals with lower economic level prefer to choose bike and bus mode, while who with higher level have a positive image of metro, and negative impression on bus. Female travelers have greater acceptance of metro-based transfer travel than male. As for travel cost, it was observed that different extents of effects on three scenarios, most significant in scenario 1, followed by scenario 3 and scenario 2. Regarding multiple types of travel time, waiting time for bus is perceived as the most important factor in scenario 1. Transfer walking time become more significant to travelers in scenario 2 and scenario 3. High punctuality of metro is the most important factor to attract commuters from car mode to combined mode.

Then, with the aim of quantify the influence degree, VOT estimation and sensitive analysis was conducted in the discussion part. We estimated the VOT of commuters in Nanjing under 3 scenarios, and VOT of waiting time, transfer walking time, walking time, and in vehicle time of combined mode. The finding of VOT estimation is consistent with the model results. Moreover, sensitive analysis of travel showed the changes of mode choice, probability with the variation of travel cost and transfer walking time.

The findings are of reference significance to provide policy recommendations, and contribute to understanding the mechanism of the combined mode.

**Author Contributions:** Study conception and design: Y.L., J.C., W.W.; data collection: W.W., J.Y.; analysis and interpretation of results: Y.L., W.W.; draft manuscript preparation: Y.L., J.Y. All authors reviewed the results and approved the final version of the manuscript.

**Funding:** This research is funded by the National Natural Science Foundation Council of China under Project No. 51638004 and the Postgraduate Research and Practical Innovation Plan of Jiangsu under Project No. KYCX18_0135.

**Conflicts of Interest:** The authors declare no conflict of interest.

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
