# Peer review of "Typical Combined Travel Mode Choice Utility Model in Multimodal Transportation Network"

_sustainability, doi:10.3390/su11020549_

Round 1

Reviewer 1 Report

The paper is aimed to explore mechanism of combined travel mode choice in multimodal networks under short, middle and long travel distance scenarios. A nested logit model is proposed for analysing travel mode choice and value of time, and sensitive analysis are conducted to describe the features of combined travel mode choice. The authors used a well know methodology, without adding any element of novelty and originality. However, the paper is well written, the survey well designed and the study case refers to a not very well explored context. So, I think that the paper can be published for spreading the obtained experimental results among pratictionnaire and researchers. The added value of the proposed research can be view also in exploring the differences in travel mode choice when short, middle or long travel distance are covered.

Generally talking, the paper is well written, but a revision of the text is necessary in order to avoid several grammatical and typing mistakes. In the following, a little bit of corrections which should be made.

In the abstract, keyword and so on, “time of value” represent VOT, that is “value of time”.

Page 2, rows 23-26: consider revise.

Page 3, rows 15-16: repeated words (transport networks and transportation network). Row 20: mainly.

Sub-section 2.2-survey design is too schematic. I suggest to better introduce the information on the way for obtaining survey design and discuss it more properly.

In section 2.3 the design of the SP survey should be better explained, by describing the factors and the level of variation of each factor generating each scenario, the considered experiments and so on.

In figure 6, the mode alternatives B+R and P+R are not well introduced. I suggest to use the terms “park and ride” and “bike and ride”.

Section 4.2 present results analysis in a very schematic manner. I suggest to the authors to discuss it more propelry. Also in 5.2 the results should be introduced in a less schematic manner.

VOT formula has to be numbered.

At page 10 please verify the sentence “with formula [21]”.

Page 10, row 24: “combined” is maybe “by combining”.

Author Response

Author's notes is attached in coverletter1.

Reviewer 2 Report

The paper is interesting and well structured.

However some elements need to be adequately stated and checked.

In particular:

- Choices:

Surveying sites. Why did the authors choose the ones presented in the paper? WHat is their significance in comparison to other areas of the city/region?

- sample. How did the author choose their sample for the questionnaire? What is the logic behind the choice of the sample?

- Is it possibile to see the survey form? How was it distributed to respondant? 

- some mispellings are present, as well as repetitions. A grammar check is needed

Some doubts remain over the design of the research itself. 

The results obtained apper very much as 'common sense' replies and devisable by simple observation rather than after a complex and articulated, though scientifically sound, application of a methodology.

THe choice of the sample, location and design of the quesions appear already as possible elements do drive the conclusions that, as said above, appear quite obvious.

Also the literature review appears as to be integrated, containing mainly a set of quite regional applications of the theory mentioned and applied.

Another element deals with the lack of reference to more recent, social-based means of mobility (i.e., uber, blablacar, etc.)

Author Response

Author's notes is attached in coverletter2.

Reviewer 3 Report

The paper is written in a very difficult to follow and confusing way. Before I can review it's content  I would like to see  this paper proofread and rewritten accordingly.

Author Response

We are sorry for our ways to write the paper. We have already revised the manuscript according to other 2 reviewers’ comments. We will be appreciating to receive your suggestions to make further promotion.

Round 2

Reviewer 2 Report

The authors followed the suggestions of the reviewers so the paper can be considered publishable.

Author Response

Thank you for your approval.

Reviewer 3 Report

Abstract: Abstract is confusing.

1.       We need to know what surveys authors talk about.  

2.       Data are plural

3.       “Based on 4170 sample of travel mode choice” – rewrite

4.       “travel time is considered among the three scenarios” – what does it bring into the sentence?

5.       “Transfer walking time and waiting time will decrease the willingness to choose bike to metro, while transfer time and travel distance factors reduce the utility of bus to metro and car to the metro.” – rewrite

6.       “Moreover, the traveler is more sensitive to the decrease of the transfer time than increase” – what time?

7.        It does not mean anything: “The value of time analysis shows that middle distance scenario as the highest value of it.”

Introduction:

1.       “The”s are added in weird places. Paper has to be proofread by a native speaker. It still reads pretty bad.

2.       Yang and Li not li

3.       Last paragraph of page one needs to be rewritten.

Data Acquisition:

1.       “Firstly, the redundant information of the questionnaire may due to the correlation between variables.” – what do you mean?

2.       What is a SP survey? Add the full name of it  in 2.1

3.        Survey design reads more as a table rather than a proper paragraph of text.  Either make it as a table or rewrite.

4.       “According to the rules that each travel mode is suitable for the distance, each scenario contains  means of transport that may generate travel competition [17,18]” – explain – rewrite

5.       Create a scheme for scenarios.  And rewrite this part of the text in the manuscript.

6.       What does discrepancy mean here? What does it bring to the table 1 ?

7.       Waiting time of bike – what does it mean?  How can one wait for a bike?

8.       Survey cannot be located. First paragraph of 2.3  needs to be rewritten.

9.       The second paragraph  of 2.3  -  needs to be rewritten.

10.    Text  under Figure 1:  questionnaire data were filled through  = why not just 644 questionnaires were filled.

11.   Why were 589  retained randomly?

12.    You do not start a sentence from AND

13.   Table 2. – you need to say what the values mean -  it is  hard to say whether these are percentages.

14.   “It can be seen that the income distribution is generally reasonable.” – what do you mean?

15.   Page 6 does not read well.

16.   Equation 1 needs explanation. What U, V and E are as well as I and j.

17.   What is j1 and jq in  Figure 3?

Model estimation and Results:

1.       Not sure whether you can say: conditional logistic regression FUNCTION.  I do not think you need references for it either.

2.       You introduce objective and subjective travel variables – have you even mentioned them before? Where did they come from?  This section is badly written and confusing.  It has to be reworked.

Discussion:

It has to be proofread and written in a less report way.  After reading this paper I do not know what new it brings to the literature. I do not think it is interesting because of the way it is written.

Conclusions:

They have to be improved.

Author Response

The responses are presented in coverletter3.

Round 3

Reviewer 3 Report

Dear Authors, 

Thank you for your corrections.  I still think it requires extensive proofreading. 

The descriptions often read as they would be from a school report not a journal paper. 

With regards